# Density of states in neural networks:
# an in-depth exploration of learning in parameter space

**Margherita Mele**  *margherita.mele@unitn.it*
*Physics Department, University of Trento, via Sommarive, 14 I-38123 Trento, Italy*
*INFN-TIFPA, Trento Institute for Fundamental Physics and Applications, I-38123 Trento, Italy*

**Roberto Menichetti**  *roberto.menichetti@unitn.it*
*Physics Department, University of Trento, via Sommarive, 14 I-38123 Trento, Italy*
*INFN-TIFPA, Trento Institute for Fundamental Physics and Applications, I-38123 Trento, Italy*

**Alessandro Ingrosso**  *alessandro.ingrosso@donders.ru.nl*
*Donders Institute for Brain, Cognition and Behaviour, Radboud University, Nijmegen, The Netherlands*

**Raffaello Potestio**  *raffaello.potestio@unitn.it*
*Physics Department, University of Trento, via Sommarive, 14 I-38123 Trento, Italy*
*INFN-TIFPA, Trento Institute for Fundamental Physics and Applications, I-38123 Trento, Italy*

**Reviewed on OpenReview:** *https://openreview.net/forum?id=BLDtWlFKhn*

## Abstract

Learning in neural networks critically hinges on the intricate geometry of the loss landscape associated with a given task. Traditionally, most research has focused on finding specific weight configurations that minimize the loss. In this work, born from the cross-fertilization of machine learning and theoretical soft matter physics, we introduce a novel approach to examine the weight space across all loss values. Employing the Wang-Landau enhanced sampling algorithm, we explore the neural network density of states – the number of network parameter configurations that produce a given loss value – and analyze how it depends on specific features of the training set. Using both real-world and synthetic data, we quantitatively elucidate the relation between data structure and network density of states across different sizes and depths of binary-state networks. This work presents and illustrates a novel, informative analysis method that aims at paving the way for a better understanding of the interplay between structured data and the networks that process, learn, and generate them.

## 1 Introduction

The history of machine learning is deeply intertwined with that of modern statistical mechanics. Since the early days of neural networks, both algorithm development and theoretical investigation have leveraged concepts and tools derived from the equilibrium statistical physics of disordered systems.

A prominent example is Gardner's pioneering work (Gardner & Derrida, 1988; Gardner, 1988), which framed the problem of learning the weights of a perceptron — the simplest single-layer neural network — as that of an equilibrium system in the presence of disorder induced by random inputs and outputs. Virtually everything that followed, from optimization techniques for training a network to the understanding of its generalization capabilities, has been addressed by the statistical physics community using the Gardner framework.

This long-lasting approach has been highly successful in characterizing the ability of simple neural network architectures to both memorize random input-output associations and generalize from previous examples (Gardner & Derrida, 1989; Seung et al., 1992; Watkin et al., 1993; Sompolinsky et al., 1990). Extensions of the formalism used to study metastable states in complex systems (Franz & Parisi, 1995) can also connect the local geometry of the loss landscape (particularly its flatness) to the generalization abilities of solutions found by actual learning procedures (Baldassi et al., 2015; 2016b; 2021), even when the dynamics of such algorithms does not necessarily satisfy detailed balance (Baldassi et al., 2016a).

The theoretical investigation of neural networks has greatly benefited from the mathematical toolbox developed in the context of spin glass theory (Mézard et al., 1987; Engel & Van den Broeck, 2001; Charbonneau et al., 2023). These methods, however, classically relied on simplifying assumptions such as the thermodynamic limit or well-defined, mathematically tractable input data, generated as independent, identically distributed (i.i.d.) variables. Thanks to these simplifications, researchers obtained a plethora of exact or approximate results that shed light on the inner workings of neural networks. A strong limitation, however, is the fact that real data are typically finite-sized and structured, thus defying the founding assumptions upon which those results are built.

More recently, novel approaches have been developed that explicitly incorporate data structure into the theory of deep neural networks, e.g., by using the second-order moments of input data (Opper & Winther, 2001; Shinzato & Kabashima, 2008; Ingrosso, 2021), or employing tasks involving effective Gaussian models and mixtures of Gaussians (Gerace et al., 2024; Mignacco et al., 2020; Gerace et al., 2021; Loureiro et al., 2024a;b; 2023). Recent work has studied the importance of higher-order input statistics for feature learning in neural networks (Ingrosso & Goldt, 2022), but the impact of such properties on the dynamics of gradient-based learning, and more generally on network behavior, is only starting to be investigated systematically (Fischer et al., 2022; Refinetti et al., 2023; Belrose et al., 2024).

The approaches mentioned thus far very often focus on the *solvability* of a learning problem, that is, the identification of those configurations of network weights that minimize the loss function. Phrasing this task in physical terms, one explores a weight configuration space equipped with a canonical probability measure, and investigates the properties of the ground states by varying the number and features of the inputs and controlling for their statistics.

In summary, a great deal of work has been carried out insofar that mainly assumes infinite, unstructured input data and focuses on a tiny fraction of the weight configuration space – that is, the part that minimizes the loss; in comparison, much less has been done to understand how *finite-sized* networks behave in the presence of structured inputs. A large body of information might nonetheless be retrieved through the study of such networks not only in terms of their solutions but also investigating the learning task globally in configuration space, by analyzing how the entire loss spectrum, defined on all possible weight configurations, is affected by a given task.

In this work, we address this problem by introducing a method to efficiently explore the entire density of states (DoS) of the loss function of a neural network. We achieve this by leveraging the Wang-Landau (WL) algorithm, a powerful enhanced sampling method used to compute the density of states of physical systems and access exponentially suppressed regions of the configuration space (Wang & Landau, 2001a;b; Yin & Landau, 2012; Vogel et al., 2014; Farris & Landau, 2021). By adaptively estimating the microcanonical entropy of the loss (that is, the logarithm of the density of states), the WL approach allows us to uniformly explore the loss spectrum, enabling a thermodynamic characterization of a neural network and its behavior for a given training dataset.

We show how the statistics and geometrical structure of realistic input data have a direct and measurable impact on the density of states of the loss function; we then highlight how specific properties can be reproduced by simple synthetic datasets explicitly controlled by few intuitive parameters, and thereby rationalize the relationship between data structure and loss density of states. In perspective, we propose that the approach here illustrated can be employed in a twofold manner, either to analyze large datasets of annotated instances, or to design networks to perform better on particular problems.

The paper is organized as follows. In the Methods section, we introduce the problem of learning in single- and one-hidden-layer neural networks focusing on the case of discrete synaptic weights, and describe the dataset used in our simulations; we then provide a brief introduction to Wang-Landau sampling. In the ensuing Results section, we first use the WL technique on real datasets, and then introduce a number of simpler synthetic classification problems. In the Discussion and Conclusions section, we discuss our findings and highlight potential generalizations of the approach.

## 2 Methods

### 2.1 Neural Network Architectures

A single-layer binary neural network, also known as perceptron, operates by constructing a hyperplane within the input space to achieve classification. Mathematically, it implements a mapping from an $N$-dimensional input vector $\xi \in \mathbb{R}^N$ to a binary output $o$ using the function $o(\xi; W) = \text{sgn}(W \cdot \xi)$, where $W$ is a (synaptic) weight vector.

In this work, we focus on networks with binary weights $W \in \{-1, 1\}^N$. Given a set of $P = \alpha N$ patterns $\{\xi^\mu\}_1^P$ and their corresponding labels $\sigma^\mu \in \{-1, 1\}^N$, the perceptron learning problem consists in finding an appropriate weight vector $\hat{W}$ satisfying $o(\hat{W}, \xi^\mu) = \sigma^\mu$ for all $\mu$, i.e. a perceptron that correctly classifies its inputs according to the prescribed labels.

The first step in formulating a statistical mechanical theory for perceptron learning is to define an energy function $E_W = E(W) = \sum_{\mu=1}^P \Theta(-\sigma^\mu W \cdot \xi^\mu)$, where $\Theta(x)$ is the Heaviside step function, counting the number of incorrect input-output associations produced by a perceptron with a given weight vector $W$.

Here, the term *energy* is borrowed from the analogy to physical spin systems, where each spin can be in one of two states (up or down), analogous to binary weights $W_i \in \{-1, 1\}$ in our neural network. The energy in these systems represents the degree of misalignment among the spins. Similarly, in our model the energy represents the number of misclassifications, and minimizing this energy corresponds to reducing these classification errors. A perfect configuration - where all patterns are correctly classified - corresponds to the "ground state" of the system, akin to achieving perfect alignment in a ferromagnetic spin system. Furthermore, due to the binary nature of the weights and the use of the sign activation function, the energy satisfies the relation $E(W) = P - E(\tilde{W})$ for any set of weights $W$ and its negation $\tilde{W} = -W$. Consequently, the amount of states for energy levels equidistant from the center of the spectrum will be identical by construction.

The learning objective can be formalized as that of finding a ground state configuration, or *solution*, i.e. a weight vector $\hat{W}$ such that the energy $E(\hat{W})$ is minimized. In a simplified scenario with random inputs and outputs, the probability of finding a zero energy solution decreases as the number of input patterns increases, i.e. the entropy of weight vectors with zero energy approaches zero as $\alpha = P/N$ grows up to a *critical capacity*. In such a setting, the application of replica and cavity methods borrowed from the statistical mechanics of spin glasses has been central in studying the storage capacity and generalization properties of neural networks (Gardner & Derrida, 1988; Krauth & Mezard, 1987; Mézard et al., 1987; Engel & Van den Broeck, 2001; Charbonneau et al., 2023).

Training a perceptron involves adjusting the weights to minimize classification errors. When constrained to binary weights, the problem becomes significantly more challenging than the continuous-weights counterpart. The use of cavity methods has been instrumental in developing powerful algorithms able to solve typical problem instances (Braunstein & Zecchina, 2006; Baldassi et al., 2007) almost up to the critical perceptron capacity.

Multi-layer architectures extend the capabilities of a perceptron to non-linear classification in input space by stacking many of these simple units in consecutive layers. In this work, we will analyze one-hidden-layer architectures, comprising an $N$-dimensional input layer, a hidden layer with $N_h$ neurons, and an $N_o$-dimensional output layer for generic multi-class classification. An input vector $\xi$ is initially mapped to a hidden layer *via* the weight matrix $W^1 \in \{-1, 1\}^{N_h \times N}$, yielding $h_k = \text{sgn}(\sum_i W_{ki}^1 \xi_i)$ for the $k$-th hidden neuron's activation. Subsequently, the hidden layer activations $h = \{h_k\}_1^{N_h}$ are linearly mapped to the

output layer using $W^2 \in \{-1, 1\}^{N_o \times N_h}$. The final output $o$ is computed by $\phi(W_2 h)$, where $\phi$ can be either the sign (when $N_o = 1$) or the argmax function, respectively for binary or multi-class classification.

For a set of $P = \alpha N$ patterns $\{\xi^\mu\}_{\mu=1}^P$ and their corresponding labels $\sigma^\mu$, we define the energy associated with weight matrices $W^1$ and $W^2$ as $E\left(W^1, W^2\right) = \sum_{\mu=1}^P \left\{1 - \delta\left[\sigma^\mu, \phi\left(W^2 \mathrm{sgn}\left(W^1 \xi^\mu\right)\right)\right]\right\}$, with $\delta$ a Kronecker delta function. Again, the learning task involves finding appropriate weight matrices $\hat{W}^1$ and $\hat{W}^2$ such that the output matches the labels for all patterns, minimizing the energy $E\left(\hat{W}^1, \hat{W}^2\right)$ to zero.

Symmetries also exist in the one-hidden-layer network, but binary and multi-class classifications must be considered separately. For binary classification, the system displays a symmetry with respect to the middle of the energetic spectrum, as it was the case for the perceptron: if both weight matrices $W^1$ and $W^2$ are simultaneously negated, the energy function remains unchanged, so that the configurations $\left(W^1, W^2\right)$ and $\left(-W^1, -W^2\right)$ produce the same energy value, $E\left(W^1, W^2\right) = E\left(-W^1, -W^2\right)$. In contrast, negating only one matrix results in an energy value that is symmetric around the middle of the energetic spectrum: $E(-W^1, W^2) = E(W^1, -W^2) = P - E(W^1, W^2)$. This symmetry arises directly from the binary nature of the sign output activation function, and it is lost when an argmax function is used in the case of multiple labels.

## 2.2 Datasets

In this work, we address the learning problem for single and one-hidden-layer networks in different scenarios involving both benchmark datasets and synthetically generated tasks.

The first scenario involves real-world datasets. The classification task is performed for various sizes of the training set $\alpha = P/N$, where $\alpha$ is gradually increased from 0.1 to 1.0 by randomly adding examples from both classes. As a result, for a given total number of inputs $P$, there might be a discrepancy in the number of examples between the two classes ($P_0 \neq P_1$): this condition is commonly referred to as an "unbalanced" dataset. By contrast, a "balanced" situation entails equally populated classes ($P_0 = P_1$). In the main text, we show results obtained on the classic MNIST dataset (LeCun et al., 1998), focusing specifically on binary classification problems involving only the first two classes (0 and 1). Additional classification problems on the MNIST dataset are reported in the supporting material (see Supplementary Section V), along with results obtained using Fashion-MNIST (Xiao et al., 2017).

In the second scenario, we use a series of approximations of an original real-world dataset that progressively capture higher-order moments of each class. In particular, we construct Gaussian clones of the datasets, by sampling inputs from a mixture of Gaussians each capturing the mean vector and covariance matrix of the images in that class. We call this configuration GM (Gaussian mixture), where inputs possess the correct mean and covariance per class but where all higher-order cumulants are zero. We also compare the results on a simplified isotropic Gaussian mixture (2isoGM), where only the mean of each class has been preserved and the pixel-wise variance of the two classes is equal to the geometric mean of the real variances.

Finally, the last two scenarios involve *synthetic* datasets. We gather hereafter some basic information on the data-generating process; all technical details can be found in the supplementary material (see Supplementary Section I). In the first case, each class represents an isotropic Gaussian distribution of points in $N$ dimensions. Two controlling parameters are employed: the inter-class separation, defined as the norm of the difference between the mean vectors of the two classes, and the angle between these two mean vectors. In the second case, we consider a perceptron learning problem with randomly generated data in a *teacher-student* setup. Each input component $\xi_i^\mu$ is drawn i.i.d. at random with probability $1/2$ in $\{-1, 1\}$, and the outputs $\sigma^\mu$ are generated using a *teacher* vector $W_T$ as $\sigma^\mu = \mathrm{sgn}(W_T \cdot \xi^\mu)$, implying that data points are linearly separable by construction as the solution $W_T$ always exists.

## 2.3 Wang Landau Algorithm

The estimation of the density of states $\Omega(E) = \sum_{\{W\}} \delta(E_W - E)$, namely the number of all internal configurations of a network with energy $E$, poses a significant challenge. In an unbiased random walk, i.e., stochastically changing the states of the network and accepting all moves regardless of the energy values

obtained, the histogram of the energies sampled along the simulation converges – up to a normalization factor – to the density of states $\Omega(E)$ in the limit of a very long walk. However, achieving such an extensive exploration over the whole energy spectrum is extremely difficult with our current computational resources, given the extremely large number of weight configurations. For example, a perceptron with 100 inputs with binary weights has $2^{100} \simeq 1.3 \times 10^{30}$ possible configurations; by assuming a previously unvisited state is sampled on average every nanosecond, it would take more than $10^{13}$ years to explore the entire configuration space.

The sampling protocol proposed by Wang and Landau (WL) offers an efficient approach to estimating the density of states of a system (Wang & Landau, 2001a;b; Shell et al., 2002; Barash et al., 2017). More specifically, in WL sampling $\Omega(E)$ is self-consistently determined through a series of *random walks in energy space*, in which the transition to a new configuration $W$ with energy $E$ is only accepted with a probability that is proportional to the reciprocal of the density of states $\Omega(E)$. Critically, the system's DoS is *a priori* unknown; the power of the WL protocol resides in its ability to reconstruct $\Omega(E)$ via a sequence $k = 0, ..., K$ of non-equilibrium Monte Carlo (MC) simulations that provide increasingly accurate approximations to the exact result.

Given the discrete spectrum of possible energy values $E$, the main ingredients of the WL iterative scheme are, respectively: (i) the *running MC estimate* of the density of states $\Omega(E)$; (ii) the running histogram of visited energy levels at iteration $k$, $H_k(E)$; and (iii) the multiplicative modification factor $f_k$ governing the overall convergence of the algorithm. For computational purposes, to fit large numbers into double precision variables it is convenient to work with the entropy $S(E) = \ln[\Omega(E)]$, further replacing the multiplicative control parameter $f_k$ with an additive one $F_k := \ln(f_k)$. Another key ingredient of WL simulations is the proposal probability governing the way in which the space of weights is explored. In essence, this rule serves as the mechanism employed by the algorithm to propose a new state starting from the current one. In this work, we relied on a combination of global and local moves (see Supplementary Section II).

Let us now describe the first step ($k = 0$) of the self-consistent scheme, the following iterations being simply repetitions of this process in which the modification factor $F_0$ is replaced by its updated version $F_k$ as detailed below. As stated previously, at the beginning of the workflow $\Omega(E)$ is *a priori* unknown. A commonly employed guess is to set $\Omega(E) = 1$, or equivalently $S(E) = 0$, for all energies $E$. The histogram is set to zero, $H_0(E) = 0$, and the control parameter to 1, $F_0 = 1$. Subsequently, a series of MC moves is performed in which consecutive transitions between two network states $W$ and $W'$, respectively, with energy $E = E_W$ and $E' = E_{W'}$, are proposed by flipping a subset of the weight vector components. Each of such moves is accepted with probability:

$$
\begin{aligned}
A(W \to W') &= \min \left\{ 1, \frac{\Omega(E)}{\Omega(E')} \right\} \\
&= \min \left\{ 1, \exp\left[-(S(E') - S(E))\right] \right\}.
\end{aligned} \tag{1}
$$

Equation 1 implies that if $\Omega(E') < \Omega(E)$, the state $W'$ with energy $E'$ is accepted; otherwise it is accepted with a probability $\Omega(E)/\Omega(E')$. We note that if the exact density of states was employed in Equation 1, the associated Markov chain would, after an initial transient, generate configurations distributed as $1/\Omega(E)$, hence resulting in a flat histogram $H(E)$. Departures from this flat histogram condition are associated with deviations of the running density of states from the true one.

If the move $W \to W'$ is accepted, the running histogram $H_0(E')$ and entropy $S(E')$ are updated according to

$$
H_0(E') = H_0(E') + 1, \tag{2}
$$
$$
S(E') = S(E') + F_0. \tag{3}
$$

In case of rejection, replace $E'$ with $E$ in the previous equations. By gradually increasing the entropy of the energy states that were previously observed and as a consequence of Equation 1, it follows that, during its

initial phases, the WL scheme tends to "push away" the sampling from already visited regions of the weight space, thus significantly boosting its exploration compared to randomly drawing network configurations. Critically, this initial random walk in energy space is continued until the histogram $H_0(E)$ becomes "flat"; at this stage, the running density of states matches the true value with accuracy proportional to $F_0 = \ln(f_0) = 1$. Subsequently, iteratively, the modification factor is reduced following the rule $F_{k+1} = F_k/2$ ($f_{k+1} = \sqrt{f_k}$), the histogram is reset to zero, $H_{k+1}(E) = 0$, and the next random walk in energy starts, the latter being interrupted when the flatness of $H_{k+1}$ is achieved. This procedure is repeated for several steps until the modification factor is smaller than a predefined final value $F_{end} = \ln f_{end}$.

The Wang-Landau (WL) scheme produces a density of states compliant with the true value – with an accuracy lower-bounded by $f_{end}$ – up to an overall multiplicative constant $C_g$. In our specific application, given the finite and predetermined boundaries of the energy range and the discrete nature of the system, it is possible to determine the global constant through simulations that cover the entire energy spectrum. More specifically, it is sufficient to ensure that the area under the DoS curve equals the number of possible configurations available to the system. For a single-layer perceptron with $N$ neurons, this is $2^N$; for a one-hidden layer network with $N$ input neurons, $N_h$ hidden neurons, and $N_o$ output neurons, this is $2^{N_h \times N + N_o \times N_h}$. A validation of the Wang-Landau sampling and a convergence time scaling analysis can be found in the supplementary material (see Supplementary Section III).

## 3 Results

Deep learning researchers are usually interested in the minimizers of the loss function. In this work, we broaden this perspective to the *whole* loss range, investigating how the number of parameter configurations associated with each value of the loss – that is, the density of states – depends on the properties of the network and its training set. Several interconnected factors, among which network architecture, activation functions, and statistical structure of the input, play a critical role in shaping the loss DoS. Rationalizing this interplay bears twofold consequences: on the one hand, it provides insight into the properties of the dataset itself; on the other hand, it can help us in understanding the inner working of task-optimized neural networks. In the following, we report the results of our work in one of these directions, our main goal being to *elucidate how data structure impacts the density of states*.

In the first part of this section, we apply our method to a benchmark dataset commonly employed in supervised learning. We examine both a perceptron and a one-hidden layer architecture solving classification tasks. A drawback in the theoretical analysis of real-world datasets is that the probability distribution underlying the input data is generally not known: to make sense of our results, we thus turn to analyze synthetic datasets whose generative model is controllable with a small set of parameters, recovering the same phenomenology observed in the case of realistic input data.

### 3.1 DoS analysis of benchmark tasks

We start off addressing the problem of computing the DoS for simple networks applied to both random and structured input data. We analyze both a binary, single-layer (SL) perceptron and a network with one hidden layer with varying size. The latter architecture is employed for binary classification as well as multi-class labeling.

Figure 1 shows the DoS for the aforementioned architectures (panel (a)) and classification tasks, demonstrating how various factors modify its properties. Panels (b) and (c) show the DoS in perceptron solving a binary classification task on both random (Figure 1.b) and real data ((Figure 1.c). We also show the results obtained in a one-hidden layer neural network with an increasing number of neurons $N_h$ in the hidden layer, for binary (Figure 1.d) and multi-class classification on the MNIST dataset (Figure 1.e)

Panels (b), (c), and (d) illustrate that, in the case of binary classification, the DoS curves are symmetric with respect to the mid-energy value ($E = 0.5$) irrespective of the network architecture. As anticipated, the symmetry is induced by the sign activation function in conjunction with the binary nature of the weights, i.e. $W = \{-1, 1\}^N$. Such symmetry is lost in the presence of multi-class classification with an argmax function

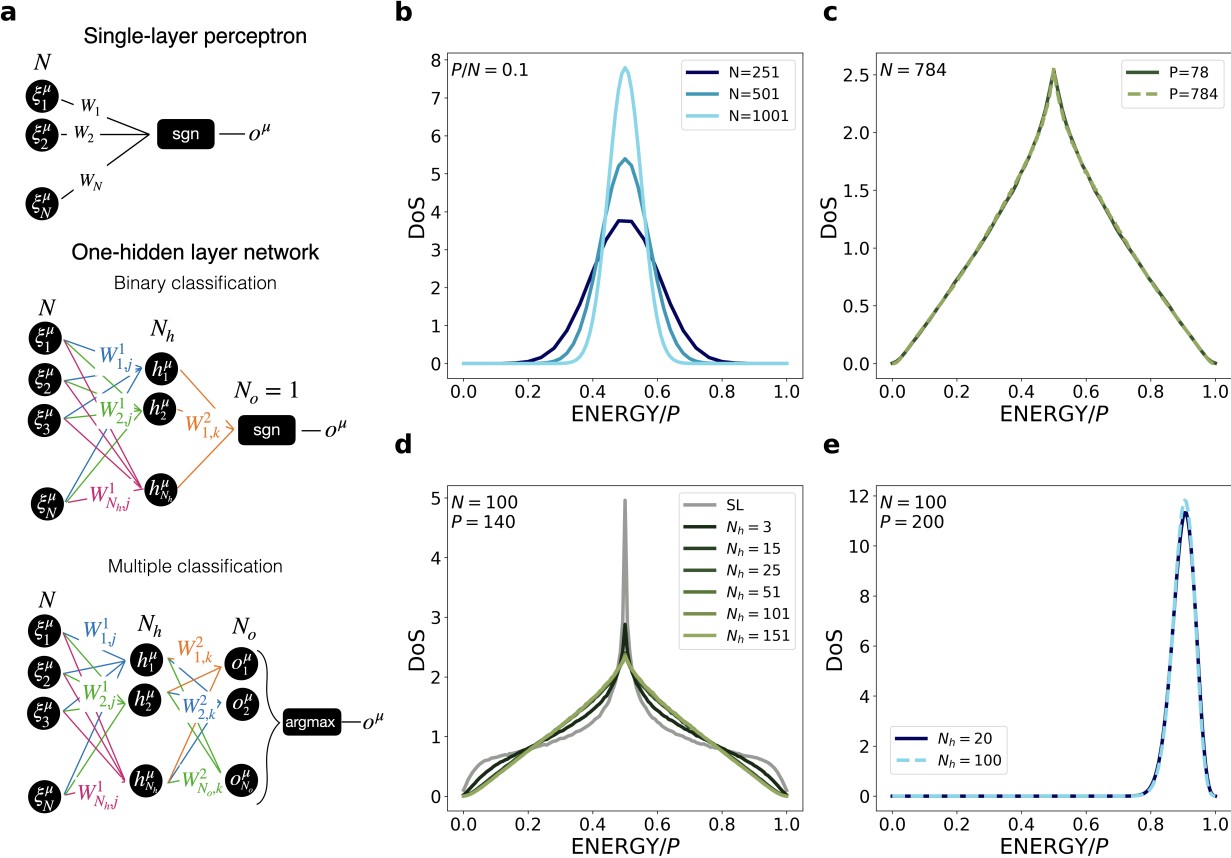

Figure 1: Comparative analysis of the density of states (DoS) in classification tasks of random or structured data across different network architectures and input dimensions. **(a)** Schematic representations of the network architectures: single-layer perceptron (top), and one-hidden layer network for both binary classification (middle) and multiple classifications (bottom), illustrating the input layer ($N$), hidden layer ($N_h$), and output layer ($N_o$), along with their respective weight connections. **(b)** Density of states for binary classification of random input data, in teacher-student setup, using a single-layer perceptron. **(c)** Density of states for binary classification of the first two classes of the MNIST dataset with $N = 784$ input dimensions by single-layer perceptron, illustrating results for different learning complexities ($P/N = 1.0$ and $P/N = 0.1$). **(d)** Density of states for binary classification of the first two classes of the MNIST dataset with $N = 100$ input dimensions, with a fixed data size of $P = 140$ and varying network architectures, including a single-layer (SL) perceptron and one-hidden layer network with different numbers of hidden layer neurons ($N_h$). **(e)** Density of states for multi-class classification of the MNIST dataset with $N = 100$ input dimensions, showing results for varying numbers of hidden neurons ($N_h$) in the one-hidden layer network.

(see Figure 1.e), where instead the DoS peaks at a value of the energy corresponding to the typical error $1 - 1/C$ obtained when guessing $C$ possible classes with uniform probability.

We now compare the characteristics of the DoS curves for binary classification *via* a perceptron in two distinct scenarios: i.i.d. data labeled by a teacher network (Figure 1.b) and structured data, exemplified by the classification of the first two MNIST classes (Figure 1.c). Both curves exhibit a maximum at the central energy value ($E = 0.5$), indicating that most network configurations incorrectly label half of the input data. However, the shape of the distribution is markedly affected by the type of data used in the binary classification. For i.i.d. data, the density of states follows a Gaussian distribution centered at 0.5 (see Supplementary Section IV). For structured data, the DoS decreases almost linearly away from the center of the energy range.

The non-Gaussian shape of the DoS is maintained even when the network architecture is modified by adding a hidden layer (Figure 1.c). Specifically, we used a one-hidden layer network with varying inner layer sizes to perform binary classification of the first two MNIST classes (reshaped into $N = 100$ dimensions, i.e., $10 \times 10$ instead of the original $28 \times 28$). We note that the general properties of the curve, symmetries and shape, are preserved. The DoS obtained with a perceptron, taken as a reference, is different from the one obtained with a dataset of larger-sized images; in particular, this curve has a sharper central peak, from which a slower decrease is observed while moving towards lower and higher energy values. This notwithstanding, as the number of hidden layers is increased, the DoS curve gradually converges to a shape qualitatively much more similar to the one observed in the case of Figure 1.b (SL perceptron, MINST dataset with images of size $28 \times 28$).

Taken together, these findings suggest that the shape of the DoS is influenced by the architecture of the network as well as by the structure of the input data, in a nontrivial interplay between the two. In order to better understand the origin of specific properties of the DoS, we thus modulated some features of the underlying data and investigated the relationship these bear with the shape of the resulting curve.

Firstly, we studied the effect of class imbalance on MNIST dataset, i.e. the discrepancy between the number of data points $\{P_0, P_1\}$ in the two classes, while keeping the total number of points $P = P_1 + P_0$ fixed. To assess the impact of class imbalance on the dataset, the DoS curves of the binary classification were computed for different values of $P_1$, thus including scenarios where class 1 was either predominant ($P_1 > P_0$) or in the minority ($P_1 < P_0$). Figure 2.b shows that as class imbalance is increased, the DoS peak shifts away from the center of the spectrum.

Irrespective of which class is predominant, we observe the appearance of two peaks that shift from the center as the imbalance increases, due to the symmetry of the DoS curve. When $P_1$ is significantly greater than $P_0$ (green curves), the DoS shows pronounced peaks away from the center, indicating a higher concentration of states at low/high energy values. As $P_1$ approaches $P/2$ (red curve) from above, the DoS curves become more centralized, indicating a more balanced distribution of states across the energy spectrum. When $P_1$ is less than $P/2$ (and thus less than $P_0$, blue curves), a pattern similar, albeit mirrored, to the first case emerges, with the peaks moving away from the center as $P_1$ is decreased.

This behavior can be understood by noting that the two symmetrical peaks in the DoS correspond to distinct sets of solutions arising from the class imbalance. The low-energy peak represents configurations where the decision boundary optimally classifies the majority class, while the minority class contributes most of the errors (and vice versa for the high-energy peak). This duality is a natural consequence of the binary weights of the perceptron and the definition of energy, which symmetrically measures the misclassifications of the two classes.

The shifting of the peaks described above is clearly reflected in a scatter plot (Figure 2.c) that correlates the position of the low-energy maximum of the DoS curve with the degree of unbalancing, computed as by $|0.5 - P_1/P|$. The plot shows results for two distinct values of dataset size, $\alpha = 0.1$ (squares) and $\alpha = 1$ (diamonds). As the degree of imbalance increases, the peak energy shifts proportionally, thereby providing a quantifiable measure of the impact of class distribution on the density of states.

These results show how class imbalance plays a crucial role in shaping the DoS, in a way that is further dependent on the structure of the input data; most importantly, class imbalance does not change the location

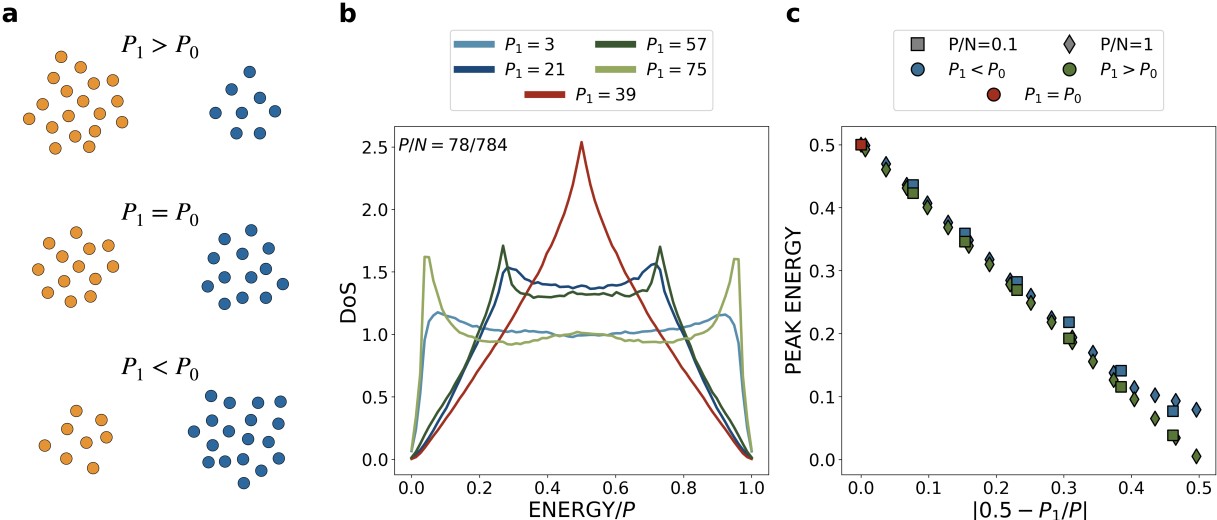

Figure 2: Density of states analysis for binary classification of MNIST digits (0 and 1) under various class imbalances. **(a)** Schematic representation of class imbalance. In the top section of the panel, $P_1 > P_0$, there are more elements in class 1 (orange) than in class 0 (blue). In the middle section, $P_1 = P_0$, the classes are balanced. In the bottom section, $P_1 < P_0$, there are fewer elements in class 1 than in class 0. The total number of points $P_1 + P_0 = P$ is fixed. **(b)** Density of states at a fixed learning complexity of $P/N = 78/784$ resulting in MNIST binary classification for different class imbalances. The legend indicates the number of elements in class 1 ($P_1$). Perfect class balance is achieved when $P_1 = P_0 = P/2 = 39$ (red curve). Larger deviations from this value indicate greater class imbalance. Blue curves represent a predominance of class 0, while green curves represent a predominance of class 1; the lighter the color the greater the class imbalance. **(c)** Peak energy values plotted against the absolute difference $|0.5 - P_1/P|$, showing the correlation between peak energy and class imbalance. This correlation does not depend on which class is predominant: blue points indicate class 0 is predominant, and green points indicate class 1 is predominant. Results are shown for two values of learning complexity: $P/N = 0.1$ (squares) and $P/N = 1.0$ (diamonds).

of the DoS peak for randomly distributed input: we observe such behavior only in the presence of structured data. This conclusion is supported by further analyzes we carried out on the MNIST and Fashion-MNIST datasets (see Supplementary Figure 4).

## 3.2 DoS analysis of synthetic tasks

To further elucidate the factors impacting the density of states, we studied simple classification problems on *synthetic* datasets, designed in such a way that a systematic study can be carried out over a small number of control parameters.

### 3.2.1 DoS in a classification problem with data from a mixture of isotropic Gaussians

Here, we focus on the DoS obtained from input data sampled from Gaussian probability densities and study its changes as we modify specific parameters of these distributions. We construct a binary classification task where the different classes consist of isotropic clouds of points in a high-dimensional space (we set $N = 41$). The distributions for each class are Gaussians with zero mean and unit variance (further details can be found in supplementary material Section I). We systematically vary the inter-class distance $\Delta\mu$, defined as the distance between the mean points of the distributions, as well as the angle $\theta$ between the mean vectors.

Figure 3.b shows the DoS curves obtained by varying the value of $\Delta\mu$, in the simple case of $\theta = 180°$. We expect that a large value of $\Delta\mu$ is associated to rather trivial classification tasks, since the two groups of points are neatly distinct and, consequently, a great number of planes in input space can successfully separate

them. In contrast, a $\Delta\mu$ close to zero corresponds to overlapping clouds: in this latter case, no plane likely exists that allows a distinction between them.

This expectation is indeed confirmed by the behaviour of the density of states: the latter is bell-shaped, with a peak at the center of the spectrum when $\Delta\mu$ is small, implying that the vast majority of network parameter configurations wrongly classify the inputs 50% of the times. As $\Delta\mu$ increases, the DoS broadens significantly: for sufficiently high values of $\Delta\mu$ the DoS develops two distinct peaks at the lowest and highest energy values, implying that most parameter sets correspond to classification planes that correctly label the data (or, alternatively, that make the wrong labeling all the times, which is equivalent to perfect labeling up to a sign).

In Figure 3.c we show the position of the DoS peaks as a function of $\Delta\mu$ ($\theta = 180°$). The orange curve represents the balanced case ($P_1 = P_0$), while the grey points correspond to scenarios with class imbalance ($P_1 \neq P_0$). In this context, class imbalance does not affect the location of the DoS peaks: the trend is indeed consistent for both balanced and unbalanced scenarios.

We then extend our analysis to a synthetic dataset where both the angle and inter-class distance are varied systematically; we focus in particular on the two different regimes of small and large inter-class distances. Results are summarized in Figure 3.d,e. For large inter-class distances (e.g. $\Delta\mu = 2$) and low angles (e.g. $\theta = 18°$), the DoS curves exhibit several properties observed in real datasets (Figure 2.b): the curves lose their typical bell shape, with the location of the maximum density being controlled by class imbalance. When $P_1 = P/2 = 30$, the density of states (red curve) displays a single peak at the center of the energy spectrum. As $P_1$ deviates from $P/2$, with larger values marked in green and smaller values in blue, the peak shifts away from the center. The peak location approaches the tail of the spectrum as $P_1$ deviates further from the balanced case.

The location of the maximum density of states reveals two distinct regimes based on the angle between the mean vectors: $\theta > 90°$ and $\theta < 90°$. In the first regime, class imbalance does not affect the peak location of the DoS curve. In contrast, for smaller angles, it plays a crucial role in determining the position of the peak, which shifts systematically towards the edges as the imbalance increases. The influence of class imbalance on the peak location is diminished when the inter-class separation is small. For small $\Delta\mu$, the maximum of the DoS remains centered in the energy spectrum, regardless of the angle between the mean vectors.

The results discussed thus far demonstrate that synthetic datasets have considerable power in replicating the behavior observed in real-world cases. This provides valuable insights in the way neural networks carry out classification tasks: particularly, in the context of binary labeling involving MNIST data, we observe that the angle between the inter-class mean vectors generally falls below 90° (see Supplementary Figure 5.a); such observation aligns with the results obtained on synthetic data, where class imbalance significantly influences the location of the peak of the DoS curves only for small angles, as it is the case for MNIST data.

### 3.2.2  DoS of a classification problem with mislabeled data

Label noise can affect the training performance and robustness of neural networks. Here we investigate how the DoS is affected by the presence of *mislabeling*, i.e., the attribution of a given data point to the wrong class, which we can easily model in a synthetic dataset by switching the label of subsets of points generated from a mixture of Gaussians.

Specifically, we analyze a setup where some input patterns in the dataset are mislabeled. Each class is an isotropic Gaussian cloud of points, with the two mean vectors identifying the classes as antiparallel ($\theta = 180°$); one cloud has a mean vector $\mathbf{m}$ and is labeled as $+1$, while the other cloud has a mean vector $-\mathbf{m}$ and is labeled as $-1$. We vary both the inter-class separation $\Delta\mu$ and the number of mislabeled elements $L$. To achieve the latter, a set of elements $L$ is chosen at random, and their labels are reversed so that points from the cloud with mean $\mathbf{m}$ are labeled as $-1$, and/or points from the cloud with mean $-\mathbf{m}$ are labeled as $+1$. The parameter $L$ ranges from 0, indicating all points are correctly labeled, to $P/2$, representing the maximum degree of disorder where half of the points are mislabeled.

Figure 4 shows that, as in previous scenarios, the peak of the DoS remains centrally located for any $L$ value when the classes are close to each other ($\Delta\mu < 1$). However, as the inter-class separation increases, the

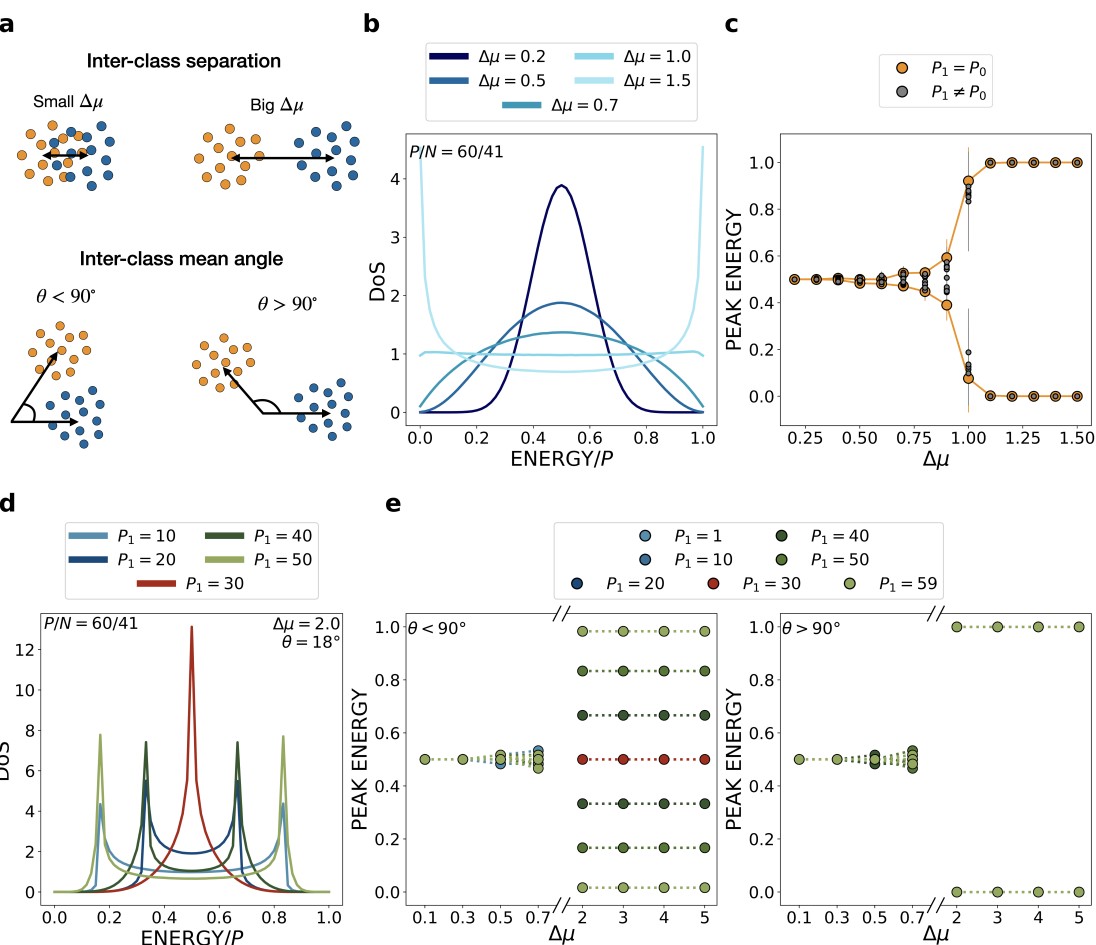

Figure 3: Density of states analysis for binary classification of *synthetic* datasets studying the impact of inter-class separation ($\Delta\mu$) and inter-class mean angle ($\theta$). The angle between the two mean vectors uniformly ranges from 180° to 18°. **(a)** Pictorial representations of inter-class separation ($\Delta\mu$) at the top and inter-class angle ($\theta$) at the bottom. **(b)** Density of States at a fixed learning complexity of $P/N = 60/41$ obtained in the binary classification for multiple inter-class separations $\Delta\mu$ and fixed inter-class angle ($\theta = 180°$) for balanced classes. The legend indicates that lighter colors correspond to larger inter-class distances. **(c)** Peak energy values plotted against the inter-class distance $\Delta\mu$ at $\theta = 180°$ for balanced classes (orange) and imbalanced classes (gray). The plotted values represent the average from 20 independent replicas of the system in the same configuration, with associated statistical deviations. **(d)** DoS at a fixed learning complexity of $P/N = 60/41$ obtained in the binary classification for large inter-class separations ($\Delta\mu = 2.0$) and small inter-class mean angle ($\theta = 18°$) for different class imbalances. The legend indicates the number of elements in class 1 ($P_1$). Perfect class balance is achieved when $P_1 = P_0 = P/2 = 30$ (red curve). Larger deviations from this value indicate greater class imbalance. Blue curves represent a predominance of class 0, while green curves represent a predominance of class 1; lighter colors indicate greater class imbalance. **(e)** Peak energy values plotted against the inter-class distance ($\Delta\mu$) for different values of class imbalance as reported in the legend. Red points denote balanced classes. Blue and green points indicate unbalanced classes with a predominance of class 0 and 1, respectively; lighter colors indicate greater imbalance. The two panels differ by the value of the inter-class mean angle, with $\theta < 90°$ on the left and $\theta > 90°$ on the right.

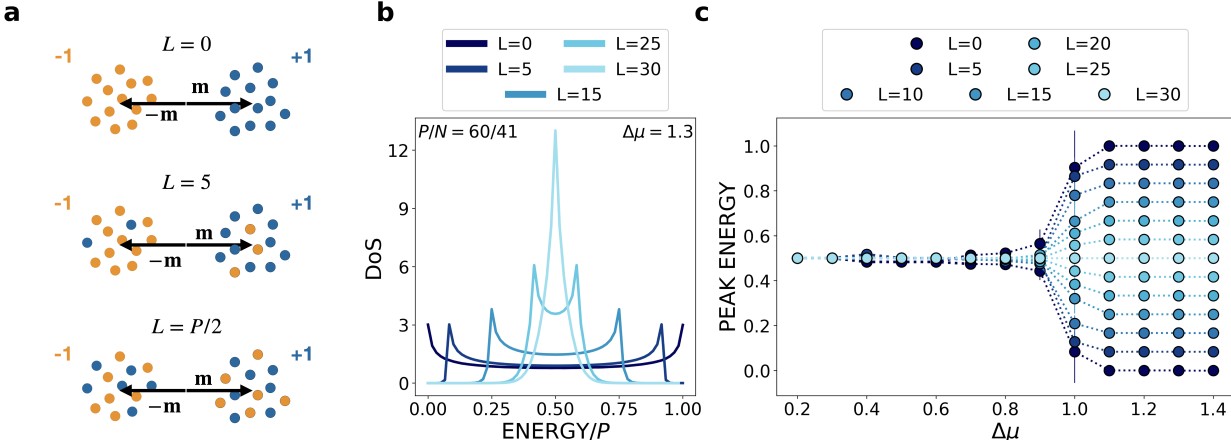

Figure 4: Density of states analysis for binary classification of *synthetic* datasets examining the impact of mislabeled elements at different inter-class separations $\Delta\mu$. **(a)** Pictorial representation of feature mislabeling for different numbers of mislabeled elements $L$. Each class is represented by an isotropic cloud of points, with one class having a mean vector **m** labeled as $+1$ (blue) and the other class with mean vector $-\mathbf{m}$ labeled as -1 (orange). The total number of points remains constant across different values of $L$. In the absence of feature noise (i.e., $L = 0$), there are an equal number of points in each class. Mislabeling is introduced by randomly selecting $L$ elements and flipping their labels. For $L = 0$, all points are correctly labeled, while for increasing $L$ (e.g., $L = 5$ and $L = P/2$), a portion of points from the cloud labeled $+1$ are relabeled as -1, and and a portion of points from the cloud labeled -1 are relabeled as $+1$, leading to label noise. **(b)** Density of states at a fixed learning complexity $P/N = 60/41$ and inter-class separation ($\Delta\mu = 1.3$), obtained in the binary classification for various numbers of mislabeled elements ($L$). The legend indicates that lighter colors correspond to a higher number of mislabeled elements (greater $L$). **(c)** Peak energy values plotted against the inter-class distance for different numbers of misclassified elements. The plotted values in the range $\Delta\mu \in [0.8, 1.3]$ represent the average from 10 independent replicas of the system in the same configuration, with associated statistical deviations.

introduction of mislabeled elements causes the peaks to shift. Notably, in the absence of mislabeling, the peaks typically appear at the tails of the spectrum for configurations with large $\Delta\mu$ and $\theta = 180°$.

As the level of mislabeling $L$ increases, the peaks move progressively toward the center of the spectrum, converging to an energy of approximately 0.5. This behavior reflects the increasing randomness of the classification task: with more mislabeled data, the network encounters a loss landscape where a large fraction of weight configurations yield similar levels of classification error, tending towards the limit of random classification. In this scenario, the density of states becomes dominated by configurations that provide no clear decision boundary between the classes. This transition aligns with the expectation that a heavily mislabeled dataset effectively loses its structure, rendering classification equivalent to chance.

This shift in peak position due to mislabeling mirrors the behavior observed with class imbalance at low values of the angles between mean vectors (Figure 3.d), highlighting the complex interplay between data characteristics and the resulting density of states. The progression of the DoS toward a central energy point emphasizes the sensitivity of neural network performance to label noise and underscores the importance of ensuring dataset quality in real-world applications (Refinetti et al., 2023; Belrose et al., 2024).

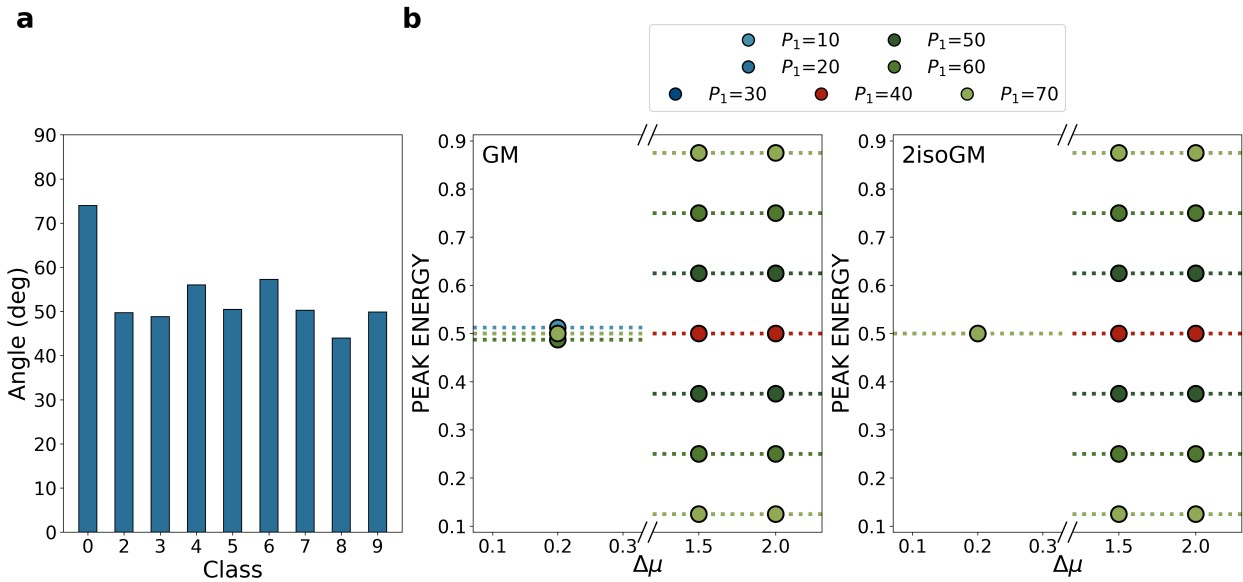

Figure 5: Density of states analysis for binary classification of Gaussian clones of MNIST data, investigating the impact of high-order statistics on DoS properties. **(a)** Angle between the mean vector of class 1 and the mean vectors of other classes in the MNIST dataset, as indicated on the x-axis. **(b)** Peak location of the DoS plotted against the inter-class separation distance $\Delta\mu$ for different values of class imbalance, as shown in the legend. Results are obtained from 9 different binary classification problems, comparing MNIST class 1 against each of the other MNIST classes. Red points indicate balanced classes, while blue and green points represent unbalanced classes, with lighter colors corresponding to greater imbalance. The two panels differ in the approximation used for the covariance matrix: Gaussian mixture (GM, left) and isotropic Gaussian mixture (2isoGM, right) clones.

### 3.2.3   DoS in a classification problem with data from Gaussian clones of real-world datasets

An interesting technique to understand the relevance of input structure on learning is the construction of a hierarchy of clones that only retain some statistical properties of the original real-world data.

In the light of this, we constructed a synthetic dataset where each class is a Gaussian clone of the original MNIST data, but where we can adjust the inter-class distances (see Supplementary Section I for further details). By doing so, we can reproduce the behavior observed in the previous sections. This is true whether we employ the real correlation matrix (GM clone) or a diagonal covariance matrix (2ISOgm). In both cases, the DoS curves show properties that closely mirror those observed with the *synthetic* datasets discussed in Section 3.2.1, particularly how the peak location is influenced by class imbalance for small angles between the mean vectors (see Figure 5). The specific structure of the covariance matrices does not seem to play a role in determining the peak location, which is mostly influenced by the difference in the mean vectors.

## 4   Discussion and conclusions

In this work, we introduced a general approach to measure the density of states in a neural network learning problem. Albeit a common practice in the field of statistical mechanics and soft matter physics (Kim et al., 2006; Stelter & Keyes, 2019; Kim et al., 2012; Tsai et al., 2007; Menichetti et al., 2021), the use of enhanced sampling techniques is only recently starting to be discovered in the context of machine learning in general, and neural networks in particular (Liu et al., 2023).

Here, for the first time to the best of our knowledge, we make use of one of these methods to thoroughly characterize the loss spectrum of simple neural architectures in a variety of learning tasks. It is crucial to note that such method, at variance with classic average-case analytical methods, can be profitably employed

on *finite-size single instances*, thus representing a complementary and promising approach to a novel class of problems and questions. Specifically, we used the Wang-Landau algorithm to elucidate the impact of several factors on the DoS, focusing on the differential effects of input statistics, linear separability, and class imbalance. The last point is particularly interesting also in view of an increasing interest in the effect of class imbalance on learning and implicit biases in deep neural networks (Francazi et al., 2024; 2023). Specifically, we addressed the effect of class imbalance on the entire spectrum of the training loss.

In this first application of our methodology we concentrated our efforts on the analysis of fully connected, one-(hidden)-layer networks; the natural generalization steps will be the study of more complex, multilayered structures, which would shed light on the impact on the DoS of weight sharing (e.g. in convolutional layers), recurrent computation (e.g. sequence-to-sequence tasks in recurrent neural networks), and multiplicative interactions (attention layers in transformer architectures). While a generalization of our technique to deeper networks is conceptually straightforward, a key issue in the usage of this protocol is, however, its scalability (see Figure 2 of the Supplementary Material for a benchmark of the convergence time): in our context, this means that its application to deep overparametrized neural networks would entail a high computational cost, deriving from the exploration of a large configuration space that grows exponentially with network size. We note that, in this work, we relied on the original, "plain vanilla" implementation of the WL algorithm due to its simplicity and suitability as a proof of concept of our approach (Wang & Landau, 2001a;b); the price to pay for algorithmic clarity and simplicity, though, is efficiency, especially on deep overparametrized tasks. This notwithstanding, the scalability challenge can be addressed through multiple strategies. Among them, we mention the parallelization across both training patterns and energy windows utilizing GPU resources (Yin & Landau, 2012; Vogel et al., 2014; Farris & Landau, 2021), as well as more advanced implementations of the basic WL algorithm making use, for example, of multi-level sampling (Vogel et al., 2013), replica exchange (Vogel et al., 2014), and energy-biasing (Valentim et al., 2015), which in the realm of soft and condensed matter physics have been shown to significantly enhance the method's sampling efficiency and scalability. In addition, (Liu et al., 2023) demonstrates the merits of using optimized WL implementations in the context of neural networks: in this work, in fact, the authors employ the method to explore the *input space* of a network, and they manage to improve efficiency and scalability by leveraging approximations such as entropy interpolation and gradient discretization. Appropriate optimizations of the WL algorithm can thus offer promising strategies to enhance the scope of our approach and its application to larger and deeper network architectures, which is the subject of further ongoing work.

Furthermore, in this work we focused on networks with discrete synaptic weights: nonetheless, in the context of soft and condensed matter physics, WL techniques have been shown to be applicable to problems in continuous space—see for instance their use in molecular simulations (Shell et al., 2002; Zhou et al., 2006; Yan & de Pablo, 2003). The extension of the WL algorithm to continuous-weight neural networks does not modify the overall conceptual framework, except that smarter techniques are needed to generate the proposal moves one employs to navigate the weight space. On the other hand, in contrast to the discrete case, such moves can rely on the differentiable nature of the loss landscape, so that strategies akin to, e.g., stochastic gradient descent, can be employed to generate new weight configurations. Further significant help in this endeavor can come from enhanced sampling techniques that are commonly employed in the field of molecular simulations (Hénin et al., 2022).

Overall, the improvements and extensions of our methodology discussed insofar open a window towards its real-world application in modern neural network models. Our method could in fact e.g. contribute to explore the mapping between the weight space and function space descriptions, in line with recent advances on linearized models in Bayesian networks (Roy et al., 2024): this could facilitate a deeper understanding of how weight configurations relate to functional outputs, complementing the insights gained from density of states analyses. Furthermore, the WL approach could provide valuable insights into selecting the best network architecture for a given problem by analyzing the DoS and the dynamics of learning algorithms as weights traverse configurations of different entropy. It is indeed well known that gradient-based algorithms do not sample the loss landscape uniformly but rather possess inductive biases (Farnia et al., 2020; Nakkiran et al., 2019; Rahaman et al., 2019) that can also affect the entire learning dynamics (Refinetti et al., 2023; Belrose et al., 2024): analyzing the dynamics of optimization algorithms from the perspective of loss density is thus an interesting avenue for future work.

## Acknowledgments

RP and AI conceived the study and proposed the method. All authors contributed to the analysis and interpretation of the data. All authors drafted the paper, reviewed the results, and approved the final version of the manuscript. The authors thank Luca Tubiana and Camilla Spreti for a critical reading and insightful comments. RP acknowledges support from ICSC - Centro Nazionale di Ricerca in HPC, Big Data and Quantum Computing, funded by the European Union under NextGenerationEU. Views and opinions expressed are however those of the author(s) only and do not necessarily reflect those of the European Union or The European Research Executive Agency. Neither the European Union nor the granting authority can be held responsible for them. RP and MM acknowledge support from Fondazione Cassa Rurale di Trento through the project SENTINEL. RP and RM acknowledge support from Fondazione CARITRO through the project COMMODORE (#20260).

## Data availability

The raw data associated with this work are freely available on GitHub at `https://github.com/potestiolab/NaNDoS`, along with a Python code to reproduce the analysis presented in the work.

## Supplementary Material

The supplementary material file provides further insight and data that support the findings of this paper. It includes:

1. Synthetic Dataset Generation: An expansion of the main text's Methods section with further technical details on how the synthetic datasets used in the study have been generated.

2. Wang-Landau (WL) Sampling Procedure: An in-depth description of the WL sampling method for estimating the density of states (DoS), including a step-by-step pseudocode.

3. Validation and Scaling Analysis: A discussion of WL sampling accuracy comparing entropy estimates with exact values, and an analysis of how convergence time scales with network size and task complexity.

4. Gaussian Fit: Analysis of the DoS distribution for binary classification using a single-layer perceptron, assessing Gaussian characteristics in specific data scenarios.

5. Fashion-MNIST Analysis: Study of how class imbalance affects the DoS in Fashion-MNIST binary classification, highlighting shifts in peak locations as imbalance varies.

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
