# Supplementary material to
## *Density of states in neural networks:*
## *an in-depth exploration of learning in parameter space*

**Margherita Mele**                                    *margherita.mele@unitn.it*
*Physics Department, University of Trento, via Sommarive, 14 I-38123 Trento, Italy*
*INFN-TIFPA, Trento Institute for Fundamental Physics and Applications, I-38123 Trento, Italy*

**Roberto Menichetti**                                *roberto.menichetti@unitn.it*
*Physics Department, University of Trento, via Sommarive, 14 I-38123 Trento, Italy*
*INFN-TIFPA, Trento Institute for Fundamental Physics and Applications, I-38123 Trento, Italy*

**Alessandro Ingrosso**                            *alessandro.ingrosso@donders.ru.nl*
*Donders Institute for Brain, Cognition and Behaviour, Radboud University, Nijmegen, The Netherlands*

**Raffaello Potestio**                                *roberto.menichetti@unitn.it*
*Physics Department, University of Trento, via Sommarive, 14 I-38123 Trento, Italy*
*INFN-TIFPA, Trento Institute for Fundamental Physics and Applications, I-38123 Trento, Italy*

**Reviewed on OpenReview:** *https://openreview.net/forum?id=BLDtWlFKhn*

## 1 Additional details on the datasets

### 1.1 Synthetic data

This section provides a comprehensive description of the *synthetic* datasets used in this study, designed to model isotropic distributions in high-dimensional space. These datasets are structured to represent two distinct classes with configurable angular separation and inter-class distances, allowing controlled manipulation of their mean vector orientations and spatial relationships. The flexibility in specifying the parameters $\lambda$ and $\Delta\mu$ enables various configurations, from parallel to orthogonal or antiparallel class centers. This setup allows for precise examination of class separability and interaction under different configurations, which is essential for assessing algorithmic performance in scenarios with high-dimensional, overlapping distributions.

The *synthetic* datasets represent isotropic distributions in $N$ dimensions, with two classes whose mean vectors are derived from orthogonal normalized vectors sampled from a univariate normal distribution. Specifically, consider two such vectors in $N$ dimensions, with components sampled from a univariate normal distribution:

$$m_c = \left\{ \frac{m_i^c}{\|m_c\|} \right\}_{i=1}^N \quad \text{with} \quad m_i^c = \mathcal{N}(0,1) \text{ and } c \in \{1,2\} \tag{1}$$

The mean vectors of the two classes are defined as follows:

$$\begin{cases} \mu_1 \equiv \Delta\mu \cdot m_1 & \text{class 1} \\ \mu_2 \equiv \Delta\mu \cdot [\lambda \cdot m_1 + (1-|\lambda|) \cdot m_2] & \text{class 2} \end{cases} \tag{2}$$

Here, $\lambda \in [-1,1]$ and $\Delta\mu \in [0,+\infty)$ are the morphing parameters controlling the angle between the two mean vectors and the inter-class separation, respectively. When $\lambda = 1$, the two vectors are parallel ($\mu_1 = \mu_2$) and the inter-class separation is zero ($\|\mu_1 - \mu_2\| = 0$). For $\lambda = 0$, the two vectors are orthogonal since $m_1 \perp m_2$,

and the inter inter-class distance is $||\mu_1 - \mu_2|| = \Delta\mu\sqrt{2}$. Finally, when $\lambda = -1$, the two mean vectors are antiparallel ($\mu_1 = -\mu_2$) and the inter-class separation is $||\mu_1 - \mu_2|| = 2\Delta\mu$. For a generic value of $\lambda$, the inter-class distance is given by:

$$||\mu_1 - \mu_2|| = \Delta\mu\sqrt{(1-\lambda)^2 + (1-|\lambda|)^2} \tag{3}$$

The angle between the mean vectors is defined by:

$$\theta = \arccos\frac{\mu_1 \cdot \mu_2}{||\mu_1||\,||\mu_2||} = \arccos\frac{\lambda}{\sqrt{\lambda^2 + (1-|\lambda|)^2}} \tag{4}$$

## 1.2 Gaussian Clones

This section aims to report the methodology used to generate *Gaussian clones*, a family of synthetic datasets designed to approximate real-world data while allowing precise control over specific statistical properties. These datasets enable systematic exploration of how features such as mean separation and covariance structure influencing DoS properties. Two distinct types of Gaussian clones were constructed: the *Gaussian Mixture (GM) clone*, which retains both the mean vector ($\mu_c$) and the full covariance matrix ($\Sigma$) of each class, and the *Isotropic Gaussian Mixture (2isoGM) clone*, which simplifies the covariance structure by preserving only the mean vectors and setting the variance ($v$) equal across classes.

For both variants, data were sampled from a multivariate Gaussian distribution $\mathcal{N}(\mu_c\Delta\mu, \Sigma)$, where $\mu_c$ is the class-specific mean vector in an $N$-dimensional space, $\Delta\mu$ is a scalar parameter controlling inter-class separation, and $\Sigma$ is the covariance matrix. The covariance matrix was defined as:

$$\Sigma_{ij} = \begin{cases} \langle x_i - \langle x_i \rangle\rangle\langle x_j - \langle x_j \rangle\rangle & \text{for GM clones,} \\ v\,\delta_{ij} & \text{for 2isoGM clones.} \end{cases}$$

Here, $\delta_{ij}$ is the Kronecker delta, and $v = \sqrt{v_1 v_2}$ is the geometric mean of the variance of the two classes. The GM clone captures the real feature correlations of the data by preserving the full covariance matrix, while the 2isoGM clone eliminates these correlations, focusing instead on isotropic variance and the effects of mean separation. This approach provides a controlled framework for studying how statistical properties of the data shape algorithmic performance.

## 2 Additional details on the Wang-Landau algorithm

This section provides an in-depth overview of the Wang-Landau (WL) sampling method implementation, focusing on the parameters and conditions essential for its convergence and efficiency. The WL algorithm is an effective approach for high-dimensional sampling, dynamically adjusting probabilities to enable uniform visitation of energy levels and accurate reconstruction of a system's density of states. We detail the selection of local and global moves, chosen probabilistically to ensure thorough exploration across energy states, as well as the histogram flatness criterion, which is critical for reliable convergence and controls the reduction of the modification factor $F$. These implementation details are vital for replicating our simulations and understanding the precision of the WL method in our study. A pseudocode summary of the algorithm is included to outline the main steps and parameter choices.

At each stage of the WL simulation, the nature of the move—local or global—is determined by a random number $c \in (0, 1)$. Specifically, if $c < 0.8$, the move is local, meaning that one of the $N$ components of the weight vector $W$ is randomly selected and flipped to generate $W'$. Conversely, when $c > 0.8$, another random number is drawn to determine the number $n \leq N$ of components to be changed. These $n$ components are then randomly selected and flipped.

The flatness condition is a crucial aspect of the WL algorithm, ensuring the uniformity of the histogram of visited energy levels. This condition is typically defined by the requirement that for each value of energy $E$, the histogram value $H_k(E)$ must not deviate significantly from the mean value $\langle H_k \rangle$. Mathematically, this is expressed as:

$$p_{flat} \times \langle H_k \rangle < H_k(E) < (2 - p_{flat}) \times \langle H_k \rangle,$$

where $p_{flat}$ is a predefined flatness parameter.

In the simulations performed in our study, we set $p_{flat}$ to 0.9. The flatness condition for the visited energy levels was checked every 800 Monte Carlo (MC) steps. The iterative simulation scheme was continued until the modification factor $F$ decreased below the predefined final value, $F_{end} = \ln(f_{end}) = 10^{-6}$.

To illustrate the implementation of the Wang-Landau algorithm in our simulations, we provide the following pseudo-code outlining the key steps of the iterative process.

```
1   Initialize:
2
3       S(E) = 0 for all E   //Initial entropy estimate for all energy levels
4       H(E) = 0 for all E   //Initial histogram for all energy levels
5       V(E) == 0 for all E  //Initial visited flag for all energy levels
6       Set modification factor F = 1
7       Initialize network state W
8       Compute initial energy E_current
9
10
11  while F > F_end:
12      //Perform n_MC Monte Carlo steps
13      for i = 1 to n_MC:
14
15          //Determine move type (local or global)
16          Generate random number c ∈ (0,1)
17
18          if c < 0.8:
19              //Local move: Flip one random component
20              Randomly select 1 component of W
21              Flip it to generate W_trial
22
```

```
23          else:
24              //Global move: Flip n < N+1 random components
25              Generate random number n ∈ [1, N]
26              Randomly select n components of W
27              Flip them to generate W_trial
28
29          Compute trial energy E_trial
30
31          if E_trial within energy range:
32              //Check if this is a new energy level (first time visited)
33              if V(E_trial) == 0:
34                  Reset H(E) = 0 for all E  //Reset histogram
35                  V(E_trial) = 1
36                  break //Restart the MC moves (break the loop)
37              end if
38
39              if S(E_current) > S(E_trial):
40                  Accept with probability exp (-(S(E_trial) - S(E_current)))
41
42              else:
43                  Accept the move
44
45              Update E_current
46              Update S(E_current) <- S(E_current) + F
47              Update H(E_current) <- H(E_current) + 1
48
49          end if
50      end for
51
52      //Check histogram flatness condition
53      Compute mean histogram value H_mean = average(H(E) for all E)
54
55      flatness_condition_met = True
56
57      for each energy level E:
58          if H(E) < p_flat * H_mean or H(E) > (2 - p_flat) * H_mean:
59              flatness_condition_met = False
60              break //Exit loop if flatness condition is not met
61
62      if flatness_condition_met:
63          Reset H(E) = 0 for all E  //Reset histogram
64          Set F = F / 2  //Halve the modification factor
65
66      end if
67  end while
68
69  Output:
70      S(E), the estimated entropy for each energy level E
```

# 3 Validation and scaling analysis

Through a rigorous application of the Wang-Landau (WL) sampling algorithm with well-defined parameters, our approach ensures a thorough exploration of the network's energy landscape, leading to precise estimations of the density of states, $\Omega(E)$, for the studied neural network.

In this section, we assess the accuracy of entropy estimations obtained via WL sampling for small neural networks. Additionally, we conduct a scaling analysis to examine how the convergence time of the algorithm grows with increasing network size and task complexity.

## 3.1 Validation of entropy estimation

To validate the accuracy of entropy estimations produced by WL sampling, we analyze small neural network architectures where an exhaustive search over the entire configuration space is feasible. This allows us to compute exact microcanonical entropy curves $S_T(E)$ and compare them with those estimated by WL sampling, denoted $S_{WL}(E)$.

Three network architectures were tested: (i) a single-layer perceptron with 15 input neurons, (ii) a one-hidden layer network with 5 input neurons and 3 hidden neurons for binary classification (one output neuron), and (iii) a one-hidden layer network for multi-class classification with 3 input neurons, 3 hidden neurons and 3 output neurons.

For each architecture, a dataset of $P$ examples was generated using an identical teacher network: $P = 15$ for the single-layer perceptron, $P = 18$ for the binary classification network, and $P = 15$ for the multi-class classification network. To evaluate consistency, we performed 500 independent WL simulations per configuration, and the resulting entropy curves are shown in Figure 1.

The figure's top panel displays the exact entropy curves (continuous red line) alongside the 500 WL-sampled curves (black points), indicating strong agreement. We quantified this agreement by computing the percentage difference between the sampled and exact curves using:

$$\text{dist}\left(S_{WL}(E), S_T(E)\right) = \frac{|S_{WL}(E) - S_T(E)|}{S_T(E)} \tag{5}$$

The mean and standard deviation of these distances, presented in the bottom panel of Figure 1, show that the average discrepancy remains consistently below $7e - 2$. This outcome reinforces the reliability of WL sampling in estimating entropy curves with high accuracy.

## 3.2 Scaling analysis of the Wang-Landau algorithm

To understand how the WL algorithm scales with system size and problem complexity, we investigated the convergence time as a function of two key variables: the number of neurons in a single-layer perceptron and the complexity of the learning task.

**Network Size Dependence**

First, we analyzed how convergence time varies with the number of neurons $N$ in a single-layer perceptron, keeping the learning complexity fixed at $\alpha = \frac{P}{N} = 1$. For each $N$ value, we generated 7 independent datasets and ran 50 WL simulations per dataset, recording average convergence times. These times, plotted in **??**, exhibit a power-law dependency, approximately $T(N) \sim 10^{-6} N^5$.

**Learning Complexity Dependence**

Next, we examined convergence time as a function of learning complexity $\alpha = \frac{P}{N}$ with $N = 100$ fixed. For each $P$ value, 7 datasets were generated, and 50 WL simulations were performed per dataset. As shown in Figure 2 convergence times follow an approximate scaling of $T(P) \sim 2 \times 10^{-6} P^5$, reflecting increased computational demands as task complexity grows with larger datasets.

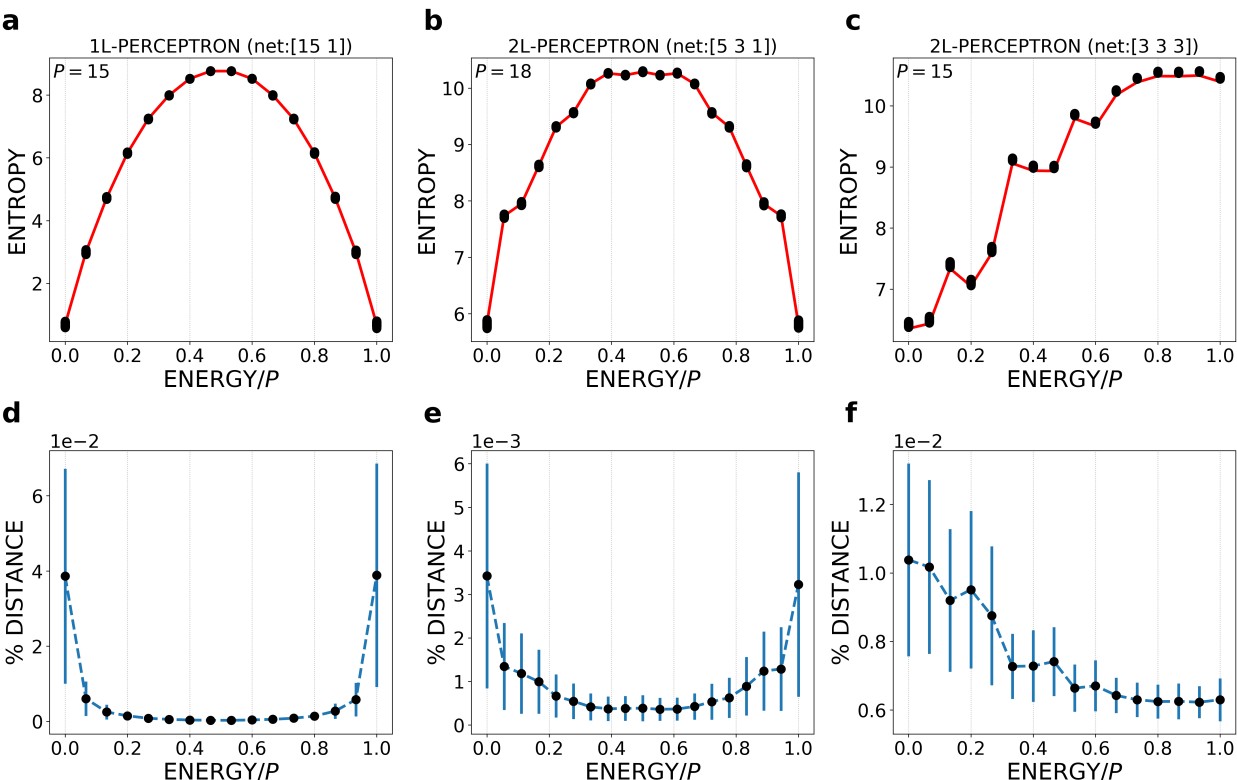

Figure 1: Comparison of the true entropy curve to the output of the Wang-Landau algorithm across different network architectures. For each architecture, 500 independent simulations are performed on the same dataset composed of P data points. The upper panels (a-c) compare the true entropy curve (red-solid line) with the sampled ones (black points). The lower panels (d-f) show the mean percentage difference between the true and sampled entropy curves, with associated errors. **(a)** Single-layer perceptron with 15 input neurons and $P = 15$. **(b)** one-hidden layer network for binary classification with 5 input neurons, 3 hidden neurons and $P = 18$. **(c)** One-hidden layer network for multi-class classification with 3 input neurons, 3 hidden neurons, 3 output neurons and $P = 15$. The corresponding lower panels (d, e, f) illustrate the mean value of the percentage difference between the true and sampled entropy curves with the associated error bars.

All simulations in this analysis were performed on a single core, which serves as a baseline for computational cost. These results could be significantly improved by leveraging multi-core processing or by implementing parallel simulations across multiple energy windows, where each window covers a specific range of the energy spectrum. This approach allows for concurrent WL sampling within each window, reducing convergence time and making the WL algorithm more suitable for large-scale or high-complexity tasks.

In summary, these scaling analyses indicate that WL sampling exhibits polynomial growth in computational cost with network size and learning complexity, both with exponents near 5. This finding provides essential insights for applying WL sampling to large neural networks or complex datasets, guiding future optimization efforts for efficient algorithm implementation.

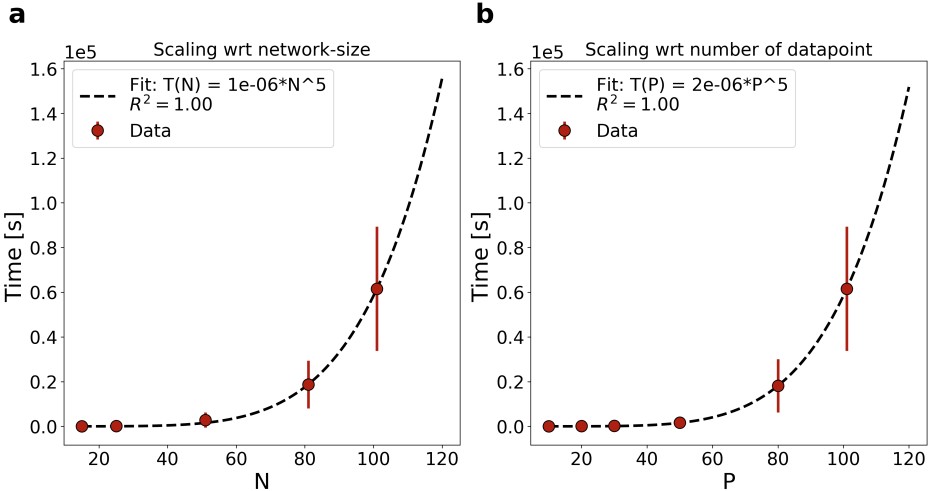

Figure 2: Convergence time of the Wang-Landau algorithm executed on a single core as a function of network size and learning complexity. **(a)** The scaling of convergence time with respect to the number of neurons $N$ in a single-layer perceptron, at a fixed learning complexity $\alpha = \frac{P}{N} = 1$. For each value of $N$, 7 independent datasets were generated, and for each dataset, 50 independent simulations were performed. The data points represent the mean convergence time, and the dashed line represents the fitted curve with $T(N) = 10^{-6} N^5$. **(b)** The convergence time as a function of the learning complexity $\alpha$, at a fixed number of neurons $N = 100$. Similarly, 7 datasets were generated for each value of $P$, and 50 independent simulations were conducted per dataset. The fitted curve is $T(P) = 2 \times 10^{-6} P^5$.

## 4 Gaussian Fit

In this subsection, we analyze the density of states (DoS) curves for binary classification using a single-layer perceptron. Our goal is to determine whether the distribution of states obtained from classifying independent and identically distributed (i.i.d.) data follows a Gaussian distribution. Figure 3 presents the DoS curves for different numbers of input neurons, $N$, along with a Gaussian fit, with the fit parameters—the standard deviation ($\sigma$) and the mean ($\mu$) of the Gaussian distribution— indicated in the upper left corner. Additionally, the goodness of fit is measured by the coefficient $R^2$, also shown in the upper left corner of each panel. The $R^2$ value indicates how well the Gaussian model represents the observed DoS, with values closer to 1 signifying a better fit. Although the density of states is not continuous in the cases studied, the DoS approaches a continuous distribution in the thermodynamic limit, where both $N$ and the number of patterns $P$ increase indefinitely while maintaining a fixed ratio. This analysis demonstrates that, under these conditions, the DoS curves closely align with a Gaussian distribution, supporting the use of Gaussian models to describe the statistical behavior of single-layer perceptrons in binary classification tasks with i.i.d. data.

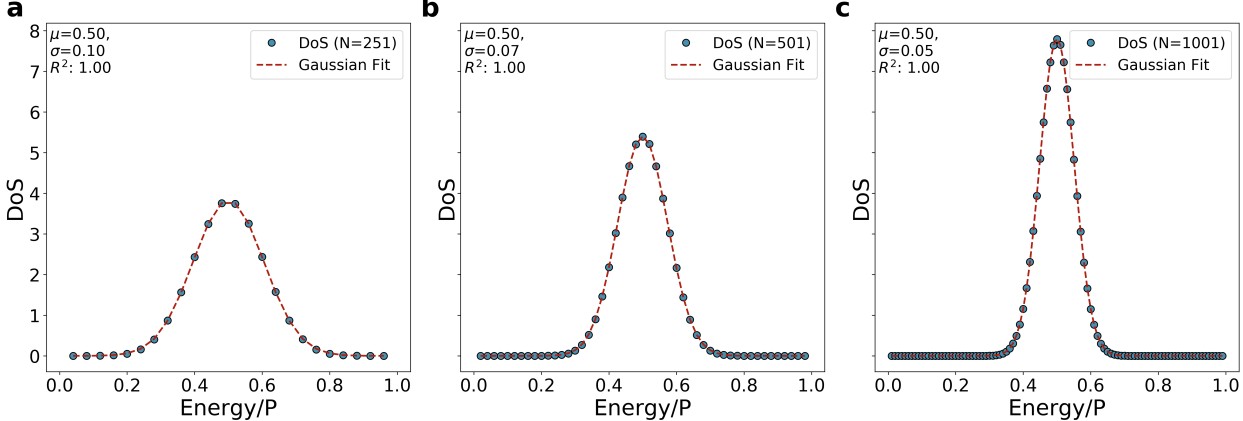

Figure 3: Density of States Curves for Binary Classification by a Single-Layer Perceptron and Gaussian Fit. Each panel presents the density of states for binary classification of random data using a single-layer perceptron with different numbers of input layer neurons, $N$ as specified in the legend. The top left corner of each plot provides the parameters of the Gaussian fit, including the standard deviation ($\sigma$) and the mean ($\mu$), as well as the coefficient of determination ($R^2$).

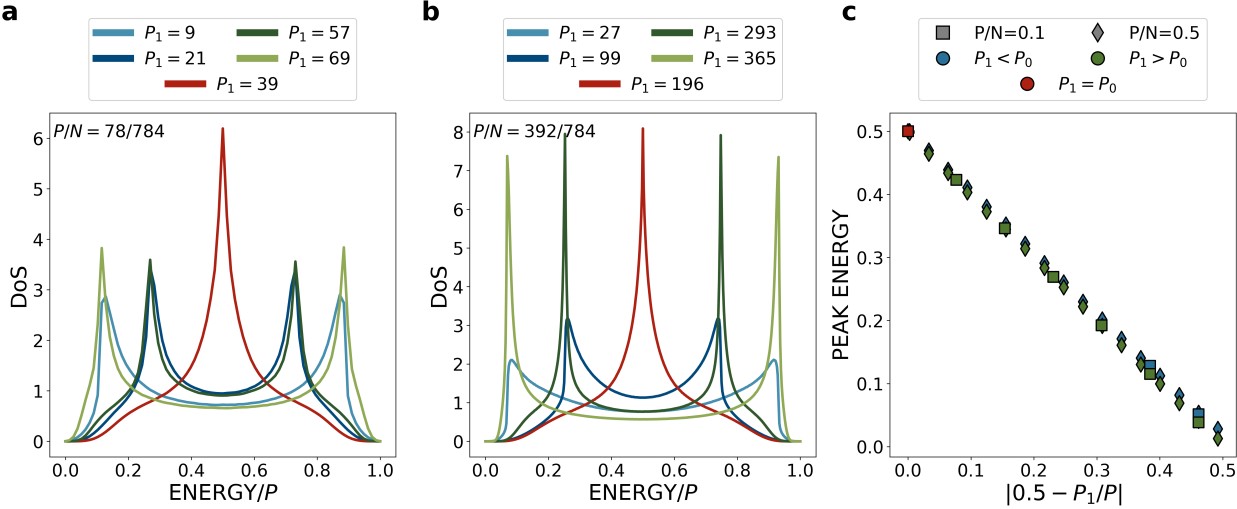

Figure 4: Density of states analysis for binary classification of FashionMNIST images (T-shirt/top and Trouser) under various class imbalances. (a) Density of states at a fixed learning complexity of $P/N = 0.1$, showing the distribution for different class imbalances. The legend indicates the number of elements in class 1 ($P_1$). Perfect class balance is achieved when $P_1 = P_0 = P/2$ (red curve). Larger deviations from this value indicate greater class imbalance. Blue curves represent a predominance of class 0, while green curves represent a predominance of class 1; the lighter the color, the greater the class imbalance. (b) Density of states at a higher learning complexity of $P/N = 0.5$, showing similar trends with varying class imbalances. (c) Peak energy values plotted against the absolute difference $|0.5 - P_1/P|$, highlighting the correlation between peak energy and class imbalance. Blue points indicate a predominance of class 0, while green points indicate a predominance of class 1. Red points represent perfect balance ($P_1 = P_0$). Results are shown for two values of learning complexity: $P/N = 0.1$ (squares) and $P/N = 0.5$ (diamonds).

## 5   Fashion-MNIST

This section provides an analysis of a subset of the FashionMNIST dataset, focusing on the binary classification of T-shirt/top and trouser images, to assess the generality of trends observed in the presence of different class imbalances. As shown in Figure 4, the results mirror those presented in the main text, reinforcing the critical role of class imbalance in modelling density of states (DoS) curves.

At lower complexity ($P/N = 0.1$), the DoS reveals a clear symmetry: as class imbalance increases, the peaks shift progressively from the center of the spectrum, regardless of which class is predominant. This pattern is maintained even at higher complexity ($P/N = 0.5$), where the separation of the peaks becomes more pronounced with imbalance. In both cases, the location of these peaks reflects the degree of imbalance, with the largest deviations from balance showing the most extreme shifts.

The correlation between peak energy and imbalance is further highlighted in the scatter plot, where the absolute difference $|0.5 - P_1/P|$ captures the relationship between class distribution and peak shift. This trend holds across different learning complexities, demonstrating the robustness of this behavior. These results on FashionMNIST corroborate our findings in the main text and emphasize that the DoS landscape is significantly influenced by structured data, with class imbalance driving non-trivial modifications in the energy spectrum.