# OpenReview forum: "Density of states in neural networks: an in-depth exploration of learning in parameter space"
_TMLR — Accepted by TMLR_

### Review · Reviewer_K1yk · 2024-12-24

**Summary Of Contributions:**

This work provides a novel method for the determination of the density of states, in terms of number of parameters which achieve a particular loss value, in a neural network model. This state density is determined by an efficient sampling method (the Wang-Landau sampling algorithm) and is applied to binary neural networks for single and multi-class classification. One benefit for this method is its application beyond simulated data to benchmark datasets. The application of this method explains and explores the impact of a number of adverse conditions including dataset bias, class scarcity, class overlap and more. Each of these conditions is then explored via the lens of density of states, allowing an understanding of the impact upon solution space.

**Audience:**

Yes

**Claims And Evidence:**

Yes

**Requested Changes:**

- Ideally, sections 3.1 and 3.2.2 would benefit with a little more discussion beyond simple observations of the results. As an example, for Section 3.1, the existence of two symmetric peaks in the DoS can be linked to the existence of sets of solutions in which the class with a larger number of samples is more or less easily classified (decision boundary is easier to find which minimises/maximises energy). Similarly for Section 3.2.2, more and more mislabelled data tends toward a random classification task with a DoS focussed upon a 0.5 energy.

- Section 3.2.3 discusses results which are in Supplementary Material, however these figures are not visible without opening up an additional document. I would request that either the relevant figures are brought into the main text, or that this section be integrated into an existing section (and not its own subsection). Currently it holds too prominent a place for the results to be unavailable in the main text.

Though this is not my area of expertise, I see this work as thought provoking and with some potential interest from the community.

**Strengths And Weaknesses:**

**Strengths**
- This paper is very well written and accessible for a researcher outside of this niche
- The results shown are extensive and consider a range of conditions under which the network solution space can be compromised due to dataset issues

**Weaknesses**
- Though benchmark datasets were used, the application to binary networks and use of MNIST limits the interpretability of the results to real-world models and tasks
- The conditions investigated in the results section were extensive, but were not always discussed in a way which could provide the reader maximum intuition. Observations of state density variation could have been combined with some more discussion.

---

> ### Author Response · Authors · 2025-01-10
> **Reply to Reviewer K1yk**
>
> We thank the Reviewer for their detailed review and constructive suggestions. Below, we address the points raised and outline the revisions we made to improve the manuscript.
>
> We acknowledge the room for improvement noted in their review. In response, we expanded the discussion to provide deeper insights and better integrate supplementary results, enhancing the clarity and accessibility of the paper.
>
> Requested Changes:
>
> - For Section 3.1, we added explanations linking the two symmetric peaks in the DoS to solution sets that differ based on class sample size and classification difficulty. This will provide more context and intuition for the observations.
> - For Section 3.2.2, we elaborated on how increasing the fraction of mislabelled data transitions the task toward random classification, resulting in a DoS centered around 0.5 loss. The added paragraph aims to clarify the implications of the results.
> - We agree that making key results more accessible is essential. We moved the relevant figure from the supplementary material into the main text. It is our hope that this will improve readability and coherence.
>
> We appreciate the Reviewer’s thoughtful feedback, which has helped us refine and strengthen the manuscript.

---

> ### Comment · Action_Editor_o6qU · 2025-02-03
> **Please submit recommendation**
>
> Dear reviewer K1yk,
>
> As the review period is coming to a close, I kindly ask you to read the authors' reply, judge whether they adequately responded to your concerns, and submit a recommendation. Please also let me know if you have any reservations that you do not want to communicate directly, or if you need any help. Thanks for your efforts!

---

### Review · Reviewer_7kEe · 2024-12-26

**Summary Of Contributions:**

This paper introduces a novel method to analyze the loss landscape of neural networks by examining the density of states (DoS) -- the number of weight configurations corresponding to each loss value. Employing the Wang-Landau enhanced sampling algorithm, the authors explore the entire loss spectrum rather than focusing solely on loss minimizers. The work reveals how properties of structured datasets influence the DoS, using both real-world datasets (e.g., MNIST) and synthetic data for validation. Key findings include the impact of network architecture, class imbalance, and data structure on the shape and symmetries of the DoS, with potential applications in understanding network behavior and guiding architecture design.

**Audience:**

Yes

**Claims And Evidence:**

Yes

**Requested Changes:**

I would appreciate more discussion on how the WL algorithm can be adapted for networks with continuous weights, as this would bridge the gap to real-world applications.

Less importantly, some empirical results demonstrating how the WL algorithm scales to larger neural networks and datasets may help convince readers the framework’s practical applicability.

**Strengths And Weaknesses:**

Strengths:

- The use of the Wang-Landau algorithm to analyze the entire loss spectrum of neural networks is novel and provides insights beyond traditional loss minimization approaches.
- The paper investigates multiple factors influencing the DoS, such as class imbalance, data structure, and network architecture, offering a thorough exploration of the topic.
- The study highlights how the statistical properties of datasets, including class balance and inter-class separation, directly affect the density of states, providing actionable insights for dataset design.
- By validating results on both real-world datasets and controlled synthetic tasks, the work bridges theoretical exploration and practical relevance. The insights into DoS offer practical applications for designing neural network architectures and understanding the dynamics of learning algorithms, particularly in structured datasets.


Weaknesses:

- The Wang-Landau algorithm is computationally expensive for large neural networks, which limits the method's immediate applicability to real-world deep learning models.
- The study primarily analyzes single-layer and one-hidden-layer networks with binary weights, which may reduce the relevance of findings to modern, deeper architectures commonly used in practice. While the method is validated on small-scale tasks, its behavior and computational feasibility for larger networks with continuous weights remain unexplored.

---

> ### Author Response · Authors · 2025-01-10
> **Reply to Reviewer 7kEe**
>
> We appreciate the Reviewer’s insightful comments, which have helped identify key areas for improvement. Below, we address each of the points raised and outline the revisions we made to the manuscript.
>
> The computational challenges of scaling the Wang-Landau algorithm and the focus on binary-weight models are important limitations. These are clearly acknowledged in the revised version of the manuscript, where we also provide suggestions for addressing them in future work.
>
> Requested Changes:
>
> - We included a discussion on potential adaptations of the Wang-Landau algorithm for continuous weights. We stress that, as it is now discussed in the revised version of the manuscript, the application of our approach to continuous-weight networks is conceptually straightforward, while some non trivial technical issues exist; these, however, can be effectively addressed making use of various strategies, some of which can be mutated from the field of soft matter physics. Indeed, it is in our plans to extend the WL framework to continuous weight spaces, and in the revised version of the paper we mention some of these forward-looking perspectives.
> - Addressing the scalability of the WL algorithm for larger networks is a non-trivial task, as the required computations scale with both the size of the network and the "complexity" of the task, as demonstrated in the Supplementary Material. In this work, we used a vanilla WL implementation both for simplicity and to provide a clear introduction to WL for the ML community; several techniques can however be leveraged, mostly known in the soft-matter community, to scale up and speed up WL sampling of systems with discrete as well as continuous degrees of freedom. Additionally, the recent work [1] referenced by Reviewer Fd7m shows that, in the context of neural networks, a gradient-based approach coupled to interpolation of the entropy function allows the WL algorithm to efficiently sample the input space to obtain the output distribution of a trained neural network.
> These enhancement techniques suggest the potential for the method to scale effectively in weight space (which typically has a quadratic dependence on the input dimension) even for complex architectures, provided that appropriate optimizations are employed. We included a discussion in the revised manuscript referencing these advancements to provide further clarity regarding the scalability of the approach and its potential applicability to larger networks. We leave the use of such approaches in larger and deeper network architectures for future work.
>
> [1] Gradient-based Wang-Landau Algorithm: A Novel Sampler for Output Distribution of Neural Networks over the Input Space, Liu et al., ICML 2023
>
> We thank the Reviewer for their valuable suggestions, which have provided clear guidance for improving the manuscript.

---

### Review · Reviewer_Fd7m · 2025-01-01

**Summary Of Contributions:**

The paper studies the density of states (DoS) of neural networks given a specific loss, which may or may not be the minimum. The paper focuses on neural networks with binary weights, and employs the Wang-Landau algorithm to draw samples while constructing estimates of the densities. Using the approach, the authors conducted experiments, investigating the loss landscape of simple neural networks. The authors highlighted the effects of different factors on the DoS, including the structure of data and network, class imbalances, inter-class statistics and mislabeling. The authors highlighted that the proposed approach is general, and is in principle applicable to larger neural networks, though the computational cost would become too large.

**Audience:**

Yes

**Broader Impact Concerns:**

The work is largely theoretical, thus not applicable.

**Claims And Evidence:**

Yes

**Requested Changes:**

The empirical studies carried out in the paper is interesting, and the claims are generally justified. However, in abstract, the authors did explicitly state "a novel, computationally efficient approach to examine the weight space across all loss values". However, as acknowledged by the authors themselves, the computational complexity becomes too large as either the network size or the problem complexity becomes too large. Thus, I think the statement is slightly misleading, and might need some rewordings.

(More like comments) It might be interesting to note that there are papers studying the function space landscape of modern neural networks, e.g. [1]. The referenced paper took a differential geometric perspective, though indeed did not study the exact densities. It might also be worth noting that previous research has also employed Wang-Landau algorithm in deep learning; I was able to find [2].

1. Reparameterization invariance in approximate Bayesian inference, Roy et al., NeurIPS 2024
2. Gradient-based Wang-Landau Algorithm: A Novel Sampler for Output Distribution of Neural Networks over the Input Space, Liu et al., ICML 2023

**Strengths And Weaknesses:**

Strengths:
1. The problem that the paper studies is interesting and quite relevant to modern deep learning.
2. Several empirical studies are carried out, highlighting how DoS varies across multiple factors.

Weaknesses:
1. The paper focuses on networks with binary weights and simple architecture, which are clearly different from modern neural networks. While I believe it is still interesting and some insights may carry over, it is a limitation.
2. The vanilla Wang-Landau sampling algorithm as presented in the paper has high computational complexity, both with respect to network size and problem complexity, thus is in general not scalable.

---

> ### Author Response · Authors · 2025-01-10
> **Reply to Reviewer Fd7m**
>
> We thank the Reviewer for their thoughtful comments and suggestions, which have provided additional perspectives for refining our manuscript. Below, we address the points raised in detail.
>
> We acknowledge the limitations of focusing on binary-weighted networks with simple architectures and the computational challenges of the Wang-Landau algorithm, and addressing them is critical in the light of extending the method to modern deep learning applications. These issues, as well as possible strategies to tackle them, are now discussed more thoroughly in the revised version of the manuscript.
>
> Requested Changes:
>
> - We agree that the phrase “computationally efficient approach" in the abstract may be misleading given the scaling challenges of our plain-vanilla implementation of the WL method. We revised this statement to clarify that the computational efficiency applies to small-scale networks and tasks, while acknowledging the challenges associated with larger networks and more complex problems.
> - In the Conclusion section we included a discussion on potential strategies to scale up and speed up WL sampling in larger and deeper architectures. In this work, we focused on simple benchmark tasks, with the aim of showing the conceptual benefit of WL as a tool to analyze the neural network loss landscape. We are currently developing more advanced implementations to deal with deeper architectures, which will be the subject of a future work (see reply to reviewer 7kEe).
> - We appreciate the suggestion to consider related studies on the function space landscape of neural networks and prior applications of the Wang-Landau algorithm in deep learning. We have incorporated a discussion of both of them in the revised manuscript.
>
> We thank the Reviewer for their constructive feedback, which has helped us enhance the clarity and contextual grounding of our manuscript.

---

> > ### Comment · Reviewer_Fd7m · 2025-01-10
> >
> > I thank the authors for their reply, which cleared my concern(s).

---

### Decision · Action_Editor_o6qU · 2025-02-10

**Recommendation:** Accept as is

**Comment:**

All the reviewers agree that the authors responded adequately to their comments and that the paper is ready for publication.

**Audience:**

All reviewers agreed that the present manuscript is of interest to the ML community.

**Claims And Evidence:**

The reviewers agree that the paper in its present form presents interesting insights into neural weights spaces. While all reviewers acknowledge that the proposed method may not scale well to contemporary architectures, they agree that the present submission is a relevant addition to the field and recommend acceptance. The presented evidence supports the main claims.